# Four-Dimensional Micro/Nanorobots via Laser Photochemical Synthesis towards the Molecular Scale

**DOI:** 10.3390/mi14091656

**Published:** 2023-08-24

**Authors:** Yufeng Tao, Liansheng Lin, Xudong Ren, Xuejiao Wang, Xia Cao, Heng Gu, Yunxia Ye, Yunpeng Ren, Zhiming Zhang

**Affiliations:** 1Institute of Micro-Nano Optoelectronics and Terahertz Technology, Jiangsu University, Zhenjiang 212013, China; 2Wuhan National Laboratory for Optoelectronics, Huazhong University of Science and Technology, Wuhan 430074, China; 3Postdoctoral Workstation, Zhejiang Chuangge Technology Co., Ltd., Zhuji 311899, China; 4School of Pharmacy, Jiangsu University, Zhenjiang 212013, China

**Keywords:** additive manufacturing, micro/nano 4D fabrication, smart materials, laser photochemistry

## Abstract

Miniaturized four-dimensional (4D) micro/nanorobots denote a forerunning technique associated with interdisciplinary applications, such as in embeddable labs-on-chip, metamaterials, tissue engineering, cell manipulation, and tiny robotics. With emerging smart interactive materials, static micro/nanoscale architectures have upgraded to the fourth dimension, evincing time-dependent shape/property mutation. Molecular-level 4D robotics promises complex sensing, self-adaption, transformation, and responsiveness to stimuli for highly valued functionalities. To precisely control 4D behaviors, current-laser-induced photochemical additive manufacturing, such as digital light projection, stereolithography, and two-photon polymerization, is pursuing high-freeform shape-reconfigurable capacities and high-resolution spatiotemporal programming strategies, which challenge multi-field sciences while offering new opportunities. Herein, this review summarizes the recent development of micro/nano 4D laser photochemical manufacturing, incorporating active materials and shape-programming strategies to provide an envisioning of these miniaturized 4D micro/nanorobots. A comparison with other chemical/physical fabricated micro/nanorobots further explains the advantages and potential usage of laser-synthesized micro/nanorobots.

## 1. Introduction

There has been growing interest in the development of dynamic shape-transforming 3D-printed structures, which is considered the next big breakthrough in additive manufacturing techniques. This innovative research aims to create 3D-printed structures that can actively change their shape over time. The term “4D printing” was first introduced by Tibbets et al. in 2013 to describe this concept of creating objects with the ability to transform their shape dynamically. Since then, researchers and engineers have been exploring various materials and design strategies to realize the potential of 4D printing and its applications across different industries. The recently emerged stimuli-responsive 4D micro/nanostructured robot (4DM/NR) is a non-negligible high-tech field at this early stage [1] and has already exhibited remarkable application prospects in aspects of micromachines [2], micro/nano-mechanics [3], embeddable robots [4,5], sensors, and bio-science [5]. These smart 4DM/NRs have become the center of many multi-disciplinary studies and are envisioned to work in various micro-environments. As pioneering studies have reported [6,7], 4DM/NRs’ small size and precise shape programming [8] offer great potential in bionics, neurorrhaphy [9], micro-optics [10], drug delivery, and highly directional locomotion [11] that are far beyond their macroscopic counterparts. Despite the tempting already achieved benefits, the high-freedom shape reconfiguration [12] of 4DM/NRs is restricted inside the laboratories due to the major challenges involving the complexity of practical motion control [13], dynamics [14], and tedious fabrication steps out of the massive application. The basic motion of 4DM/NRs, such as simple bending or expansion, is becoming common [1,2,3,4,5,6,7,8,9,10,11,12,13,14], while complex 4D programming, such as out-of-plane reversible gripping, selecting stimuli type, dynamic information encoding/transmission, and self-healing ability from physical damage, remains rare and pose a significant challenge for researchers.

To break through the technological/material limitations of implantable 4DM/NR applications, much effort has been devoted to shape-controlling strategies [15] to incorporate advanced light-triggered photochemistry [16,17] and soft-responsive materials [18]. In this respect, we have summarized and analyzed our previous work and peer research about laser-direct synthesized 4DM/NRs (see schematic in Figure 1) for review. To begin, the photochemical mechanism, which mostly relies on the photon-induced crosslinking process [19] (a photochemical reaction known as single-photon [20], two-photon [12], and multi-photon [21] polymerization) was introduced with an optical setup. To scale the robots’ volume down to micro/nano-environments [22] with improvements to intelligence, the fabrication resolution of laser synthesis has to be confined to nano-size [23] at a flexible scanning path. In parameter-tunable laser photochemical fabrication [24], the spatiotemporal programming depends on the method of modulating photons [18]. The obtained stimuli-responsive micro/nanostructure is an energy converter that reversely transforms input energy to other forms, promising geometrical reconfiguration and leading to optical, bio-electrocatalytic [25], optical, electrical, or biological responsiveness [26]. Therefore, how to evaluate and optimize the micro/nanoscale interaction with various stimuli is the second challenge, which determines the behaviors of biomimetic 4DM/NRs.

The key characteristic of active 4D materials is their stimuli-responsiveness. Currently used 4D materials include functional hydrogels [24], liquid crystal elastomers [27], responsive organic polymers [28], proteins, and their nanocomposite materials [29]. For this ultrafast photochemical polymeric reaction, its underlying molecular dynamics and material science deserve specific investigation, which intrinsically determines the fabrication resolution [12,18] and programming method. At the end of this review, we compare laser-synthesized mini-robots with robots obtained via other current chemical/biological/physical methods [30] in various fields of use to highlight the advantages of the reviewed digital photosynthesis methods.

## 2. Micro/Nanoscale Laser 4D Additive Manufacturing with Smart Materials

Accelerated by the coming era of “Industry 4.0”, the digital 3D production processes became a revolutionary development in efficiency, resolution, geometric capacity, and functionalization, incorporating new smart materials. The geometry, volume, and resolution optimization of laser photochemistry [31,32] has developed beyond the traditional aim, contributing to new highly valued 4D products; its fabrication process can be explained as a series of photochemical reactions starting from photon absorption (as seen in Figure 2a). The photon energy excites the ground-state atoms of photoinitiators or monomers [33] to a higher energy level (known as photon absorption), making the photoinitiators or monomers unleash unstable free radicals, the free radicals trigger chemical reactions to link the molecular chain of monomers into a matrix (known as crosslinking or polymerization) or reduce the metallic ions into crystals (known as reduction). The short-wavelength photons possess higher light energy than long-wavelength ones, therefore, the short-wavelength laser beam is prone to incur single-photon [34] absorption, while the photoinitiator absorbs two [35,36] or several [21,33,37] photons simultaneously to incur two/multi-photon absorption. The subsequent reactions transform the liquid precursors into a solidification state at a micro/nano resolution.

Many studies have successfully proved several stimuli-responsive smart materials for micro/nano 4D printing [25,33,37,38] (Figure 2b). Hydrogel [39,40,41,42,43] is a typical kind of functional material with a crosslinking three-dimensional network [18], and it is hydrophilic and insoluble in solvents. Especially, our recent work report an amphiphilic hydrogel containing both hydrophilic and hydrophobic groups [12], making the micro/nano 4D products respond to various stimuli rather than simple humidity. The hydrogels/polymers are also ideal soft carriers of magnetic nanoparticles [35,44,45,46], mechanics-reinforced carbon nanotubes [24], photon-sensitive fillers, drug particles, or electrically conductive dopants. Using these advantages, the cross-linked network can generate responsiveness under small loading of environmental stimuli, such as temperature, magnetic field, electric field, or multiple stimuli. The intelligent hydrogel networks usually present a mesoporous or fibrous network with unfoldable space to absorb or desorb the solvent and produce swelling or shrinkage, the recent work shows the swelling/shrinkage volume change ratio easily exceeds 200% [37,38,39,40,41,42]. Due to hydrogel’s desirable biocompatibility [47] and mechanical properties, micro/nano 4D hydrogel can be further combined with in vitro cell culture, vascular dredging [48], or be used to emulate the cargo/reorganization of organisms [49].

Besides hydrogel, shape memory polymer [14,15,50], another special material, has attracted widespread attention, as it takes the glass transition temperature as the critical temperature for phase transferring. When the ambient temperature is lower than the glass transition temperature, the shape memory polymer is in a fixed phase, with a memory structure shape having a large modulus and hardness [51]. When the temperature is higher than the glass transition temperature, its physical properties undergo significant changes, with the modulus and hardness gradually decreasing and softening, which is a reversible phase that ensures shape changes by temperature variation [52,53,54,55].

As has been previously reported, liquid crystal [56] is another stimuli-responsive substance between crystals and liquids, it possesses the fluidity of liquids and the orderliness of crystals. Micro/nanocrystals are connected to the polymer skeleton and moderately crosslinked to obtain liquid crystal elastomer [57]. The increase in environmental temperature or under the irradiation of light causes liquid crystal molecules to change from a liquid crystal state [58] to an isotropic state, resulting in thermally induced deformation. The change of orderliness of the crystals also induces tunable optical function. In the current literature, liquid crystal transforms into elastomer via polymerization, and the elastomer is both soft, rigid, and widely used in robotics and optics.

Some other prior research has demonstrated protein [59] as another bio-compatible smart substance, as its certain spatial structure is formed by the winding and folding of peptide chains composed of amino acids through dehydration and condensation [60]. In laser-induced synthesis, some functional proteins form micro/nanostructures through amino acid bonds, the amino acids carry diverse functional groups that endow proteins with the ability to respond to external stimuli as biological actuators [61]. For example, bovine serum protein has carboxyl and amino groups on its molecular chain; it can cause morphological changes in protein molecules by pH variation. 

These above-mentioned smart materials have garnered significant attention for their impressive deformability and scalability for micro/nano 4D printing. However, what we know about them is largely based on observation and laboratory experiments. Their imperfection still deserves in-depth studies for practical use. For example, devices based on hydrogel materials often suffer from relatively poor mechanical properties, limiting their practical applications. Liquid crystal elastomers tend to exhibit smaller degrees of deformation compared to other materials, and their response speeds are generally slower. The deformation of shape-memory polymers is primarily induced by changes in temperature, its simplicity in actuation mode comes at the cost of slower response speeds. In a nutshell, the innovative approaches enhancing mechanical properties, shape-morphing controllability, capabilities, and ability, need to be further explored. 

After the introduction of used smart materials, we summarize the laser systems (Figure 3a_1_,a_2_) for photochemistry. The beginning is the coherent laser source; previous researches tell us that the nonlinear laser crystals (neodymium-doped yttrium aluminum garnet or yttrium aluminate, titanium-doped sapphire, potassium dihydrogen phosphate, and so on) work with advanced modulation techniques to irradiate laser beams covering ultraviolet-to-infrared wavelengths. Optical parametric oscillators [62] hold the responsibility to tune light wavelengths based on the nonlinear four-wave mixing effect [63], the wavelength-tuning is realized by physically changing the angle between the incident pump light and the laser crystal axis [64]. 

To fabricate delicate 4D shape-reconfigurable micro/nanostructures using these lasers, a digital optical platform (Figure 3) consisting of the above-discussed light sources and modulation setup is required. As previous works reported that a green micro/nano 4D manufacturing optical platform [65] developed from the digital 3D nano printing system [66] (Figure 3a_1_,a_2_) should simultaneously possess the interactive functions of complex model analysis and slicing, spatially laser beam control and modulation ability, scanning path programming, high compatibility with multi-materials, and so on. For this aim, the optical setup of conventional micrometer-resolution lithography or projection evolves to a 4D printing nano-accuracy structure pre-designed for shape reconfiguration.

Studies [67] show that the stepsize of shape-programming is a single layer in 3D lithography or projection; it is the minimum programmable unit that needs to be designed and unevenly patterned, limiting the shape-programming resolution at its area size [68] at the micrometer scale. For standard micro/nanoscale 4D printing (as seen in Figure 3), when using a femtosecond laser direct writing setup [18], the basic programmable unit is the laser-focused voxel [69]; therefore, the shape-morphing resolution reaches the <100 nm level. In the setup with the stimulated emission depletion (STED) [70], even a resolution of <50 nm [71] becomes achievable. However, STED setup strongly depends on negative photoresists [72], limiting the functions of smart materials.

The literature review [22,25,33,34,35,36,37,62,63,64,65,66,67,68,69,70] also showcases the development of opto-mechatronics. The laser direct-writing setup adopts an orthodox “2 + 1” configuration, 2D scanning galvanometer/micromirror [73] plus 1D translation stage, advanced 3D galvanometer [74] with focus-adjusting ability, or inserting spatial phase modulators to form a “2 + 1” configuration. The recently reported rotating strategy uses the rotation freedom of the substrate to print 3D structures, also showing desirable feature sizes. By modulating a single laser beam to a point array or specific pattern, the fabrication efficiency or volume is ameliorated over one magnitude. In summary, the laser photochemical synthesis utilizes an optical setup of the digitalized laser additive manufacturing system [75,76,77,78] with precise optical components (focusing optical path, radio-frequency acoustic–optic modulator, 3D scanning system, large N.A. objective lens) to confine the photochemical reaction at the nanometric resolution, and is being developed by researchers from several fields.

## 3. Shape Programming Strategies by Ultrafine Nanostructures

Numerous current works emphasize the programming strategies of 4DM/NR biomimetic function [12,13,14,15,16,17,18,19,20,21,22,23,24,25,26,27,28,29,30,31,32,33,34,35,36,37,38,39,40,41,42,43,44,45,46,47,48,49,50,51,52], allowing the artificially made 4DM/NRs to realize high-level motions beyond nature. To spatiotemporally program reverse 4D behaviors, the shape-programming strategy is crucially important and not inferior to structural design or smart materials. The shape-programming strategies of previous studies can be categorized into three levels: the layer (Figure 4a), the microscale meta structures (biomimetic cilia with meta microstructures in Figure 4b–d), and the single nanowire (Figure 5, linewidth controlled by single laser focus). The 2D single-layer pattern at different layouts generates different swelling or shrinkage degrees and is used in light-projection fabrication. By designing the layout of each layer, the shape reconfiguration can be controlled by the light projection fabrication. In dual/multi-layer design [79,80], shape-morphing happens by using different materials, forming a soft–rigid structure, where the rigid layer works as a skeleton, and the soft layer works as a muscle [81] to generate direction bending. However, according to existing analyses [32,33,34,35,36,37,38,39,40,41,79,80,81,82,83,84,85,86,87,88,89,90,91,92,93,94,95,96,97,98,99,100,101,102,103,104,105,106,107,108,109,110], the mechanical mismatch between multiple layers is a pitfall for repeated shape-morphing in actual usage.

The latest literature indicates that these fundamental shape-programming strategies have evolved by introducing meta-mechanical structures into the multi-layers. For instance, our peer researchers built small 3D micro cage-like micropillar meta structures [82] to replace the planar soft layer along with the rigid layer, creating the unique meta-mechanical micropillar-triggered shape configuration of cilia using two-photon polymerization (Figure 4b), where the meta-mechanical micropillar cells present anisotropic shape-morphing and strong adherence to rigid layers, ensuring high-order direction in reverse photothermal shape reconfiguration, and the spatial resolution of shape-programming depends on the volume of a single micropillar at the microscale.

In the work of [82], one remarkable advantage of smart hydrogel is its ability to undergo precise and programmable light-induced transformations with exceptional spatial resolution. Researchers focus a laser beam (approximately 1 µm^2^) on either the face or edge of a cubic woodpile structure measuring 50 × 50 × 50 µm^3^ (length × width × height, Figure 4c). Upon exposure to the laser, the woodpile exhibits a fascinating response—it shrank around the laser focus area and rapidly assumed various stable 3D morphologies. This newly found dynamic behavior demonstrates the versatility and controllability of smart hydrogels in achieving complex shape transformations at microscopic scales. As seen in Figure 4d, researchers also report a 3D microclamp consisting of a circular array of six micropillars, and its functionality is demonstrated through optical micrographs taken under different light-stimulation powers in an aqueous environment. This meta-mechanical micropillar provides a much larger surface/volume ratio than the traditional multi-layer structure, making the stimuli-responsive behavior violent. Predictably, by devising meta-mechanical cells and implanting them into the soft-to-grid design, high-performance biomimetic 4DM/NRs could be realized.

In Figure 5a_0_, it can be seen that the volume of the laser focus (known as “voxel”) directly affects the scanned nanowires; the photon polymerization reaction happens only inside the “voxel”. Previous investigations [18,62,63,64,65,66,67,68,69,70] conclude that the voxel shape focused by a Gauss laser beam is an eclipse at the nanoscale and that the volume of formed nanowires (its height and linewidth) depends on the laser voxel during scanning. Accordingly, the spatial resolution of the laser-scanning-induced nanowires, patterns, and structures is determined by the specific volume of the laser voxel. To push the shape-programming resolution and capability to their limits, in our latest work, heterojunction scanning [18] is experimentally confirmed as an effective strategy, in which laser scanning maintains single-voxel resolution and introduces uneven crosslinking degrees into the monolayer (a 2D metastructure containing both hard and soft layers, Figure 5a_0_–a_3_) with different densities, directions, and positions, making the shape reconfiguration anisotropic (expansion happened at pointed direction, Figure 5b_1_–b_3_) instead of the traditional slow isotropic swelling.

Our reported experiment results have guessed that the 4DM/NRs of hands and grippers in Figure 5a_0_–a_3_ act based on the non-covalent bonding effect [12,18]. Non-covalent bonding refers to the interactions between molecules or particles that do not involve the sharing or transfer of electrons. These interactions include hydrogen bonding, van der Waals forces, electrostatic interactions, and hydrophobic interactions. To precisely harness intermolecular forces, heterojunction scanning is proposed to manipulate more complex 4D behaviors in the monolayer than a tedious multi-material multi-layer reciprocating scanning strategy, which also eliminates the mechanical mismatch during multi-layer interfacing [51,80]. 

In heterojunction scanning, the direction of shape bending/expansion is perpendicular to the laser-scanned nanowires, and the bending direction can be controlled by adjusting the laser scanning direction to realize chiral torsion, anisotropic deformation, and site-specific mutation, in which the hydrogel structures are selectively modified or altered at specific locations. Moreover, the hydrogel nanowires demonstrated spontaneous self-repairing capabilities, allowing them to recover their original shape or functionality after being damaged or deformed. This reusability makes them promising candidates for micro/nanorobotics. Beneficially, the dimension of the 1D nanowire is traceable to the 0D voxel (Figure 5a_0_–a_3_) of laser focus [81,82,83]; therefore, the monolayer or formed 2D metastructures realize nano-accuracy programming based on the spacing and linewidth of the nanowire without losing shape reconfigurable capacity, promising future molecule-level robots.

In the above-demonstrated programming strategies, digitalized programming contributes to the tempting advantages of precise spatial controllability, outperforming traditional hydrothermal, sonic, and electrochemical methods. Moreover, the micro/nanoscale volume after shape programming decreases material cost and offers a larger surficial area than those bulky ones, and the inside mesoporous [83,84] or fibrous [12,18] network naturally offers numerous diffusion channels for transporting stimuli (solvent, ions, heating, and so on), making the responsiveness violent and fast when compared to slowly swelled, bulky ones [85]. Anyway, shape programming has a huge developing space and allows for huge geometric flexibility compared to conventional chemistry for better complexity in motion control, feature size, and function integration, and there exists a huge developing space.

## 4. Self-Driven Mechanism by Various Stimuli

The driven mechanism of the 4DM/NRs has been an extensively studied field, as it is crucially important to performance controlling and improvement of 4DM/NRs. Among the existing works, the noncontact magnetic field drive (as seen in Figure 6a_1_–a_3_) is often used for the remote control of microrobots due to its strong penetration ability, long driving distance, and high-order direction [45,86,87,88,89]. Because photocurable precursor materials are doped with magnetic nanoparticles (Fe, Ni, NdFeB, etc.) [44], by rotating the outside-applied magnetic field, the micro/nanostructured filter rotates at desired angles and frequencies accordingly for the filtering modes.

This innovative microfilter enables switching between two distinct modes: filtering and passing (Figure 6a_2_,a_3_). One notable advantage of this magnetically driven rotary microfilter is its ability to perform multi-mode filtering functions. For example, it can effectively capture particles with a size of 8 µm while allowing particles measuring 2.5 µm to pass through. It also can accommodate situations where both types of particles need to be passed through without filtration. The responsive characteristic of the microfilter, enabled by magnetic control, significantly enhances its reusability. 

Numerous works have proved that temperature-dependent response [90,91] is another widely driven mechanism (Figure 6b). As known, the polymer, hydrogel, liquid crystal elastomers, or proteins generally shrink under heating conditions. Based on this, the reported woodpile photonic crystal/spiral disk shrinks to change its dimension after heating. The second driven force is the intermolecular-effect-induced dynamic force by responsive stimuli of various solvents [18]. Due to the presence of molecular interactions between the material matrix and solvent, the micro/nano 4D product exhibits significantly different swelling or shrinkage abilities with different solvents. pH stimulation responsiveness [83,91] has received widespread attention in micro/nano 4D printing due to high response speed [92] in milliseconds and excellent actuation amplitude. pH changes induce structural deformation, as demonstrated in Figure 7, the carboxyl groups in polymers/proteins undergo deionization in acidic solutions or ionization in alkaline solutions [93,94], causing expansion or contraction due to the electrostatic forces in the polymer or protein molecular chain [95,96,97].

As illustrated by Figure 7, pH responsiveness is a novel-driven mechanism for fabricating micromechanical devices with a nonuniform internal lattice density. The technique employed in this research is the femtosecond laser two-photon polymerization, which allows for the flexible design and high-precision production of 3D micro/nano devices. The increase in scanning step length from 100 nm to 200 nm led to a higher swelling ratio of BSA micro/nano blocks [96]. This can be attributed to a decrease in the cross-linking density of the processed structures. In a pH 1 solution, the swelling ratio was approximately 1.9 when the step length was 200 nm. On the other hand, in a pH 13 solution, the swelling ratio reached 265. When using a step length of 100 nm in an alkaline solution, the swelling ratio was found to be 76.7% of that observed in structures fabricated with a step length of 200 nm.

According to the current literature, light [98] is another remote noncontact drive for 4DM/NRs and is used to precisely stimulate other physicochemical effects to induce structural actuation, such as the photothermal effect, photochemical effect, and optical force. Photothermal conversion [99,100] is an effective form of energy conversion [101,102] that uses the photothermal effect of materials to induce local temperature gradients to induce structural shape changes. Some designs are implemented through the transformation of azobenzene moieties in a liquid crystalline elastomer matrix by laser direct writing to exhibit motion triggered by ultraviolet irradiation and near-infrared responsive mechanical deformation [102]. Photochemical effects are based on specific photoresponsive chemical materials, such as azobenzene or spiropyran [103] derivatives to produce photoisomerization under light stimulation, changing the polymer chain length and thus deforming the device structure. Several envisioning robotic devices with functionality have been produced from a single azobenzene-functionalized cholesteric liquid crystal elastomer. The other driven mechanism is pure optical force, which realizes the motion of an object by using the gradient force of the light field itself or the momentum of photons.

In the above-listed driven methods, each driving method comes with its own set of challenges. Magnetic responses are limited by their driving distance capabilities and raise concerns about the safety of high-intensity magnetic fields in biomedical applications. Achieving accurate control of pH and solvent responses can be challenging. Temperature responses tend to be slow. Light responses suffer from low energy conversion rates and issues with tissue transparency. To achieve widespread application scenarios in the future, an ideal actuation method for micro/nano 4D printing needs to fulfill several requirements simultaneously and should be capable of noncontact stimulation over long distances to accommodate complex working environments. Additionally, it should offer precise control and agile responses to meet the demands of intelligent integrated devices. Among previously reported driven methods [82,83,84,85,86,87,88,89,90,91,92,93,94,95,96,97,98,99,100,101,102,103], light responses hold great promise as an ideal driving and control method. This is due to their rich wavelength selectivity, precise spatial targeting capabilities, and multi-dimensional regulatory abilities. Light responses offer the potential for achieving complex movements and deformations in micro/nano structures with high precision and efficiency.

## 5. Prospective Future and Applications of 4DM/NRs

After the introduction of versatile shape-programming strategies and the driven mechanisms of the 4DM/NRs, determining a method to predict and apply 4DM/NRs has grasped our attention [104,105,106,107,108,109,110,111,112,113,114,115,116,117,118,119,120]. To enhance the control over shape transformations while minimizing response time, researchers have been exploring nanofabrication approaches with sophisticated designs. A groundbreaking four-dimensional microprinting strategy is firstly introduced for constructing 3D-to-3D shape-morphing micromachines [104] in a single-material–single-step mode. By utilizing direct laser writing, the researchers can spatially distribute heterogeneous stimulus-responsive hydrogels into arbitrary 3D shapes with sub-micrometer features. The material properties of the hydrogels, including crosslinking densities, stiffnesses, and swelling/shrinking degrees, can be precisely modulated by programming the exposure dosage of femtosecond laser pulses.

Taking use of proper structural design and numerical prediction, more complex biomimetic shape reorganization (machine-level mutation [104], untethered jumping [105], directional swimming [106,107], and so on) becomes realizable, and response time is greatly shortened with improved performance. These micromachines exhibit rapid, precise, and reversible 3D-to-3D shape transformations in response to multiple external stimuli. To predict the shape reconfiguration trend for complex structures, researchers have increasingly employed finite element analysis (FEA) simulation [18,104]. This computational method allows them to simulate, predict, and analyze the behavior of 4DM/NRs during shape transformation. By combining FEA with advanced manufacturing techniques like four-dimensional direct writing (4DLW), researchers can accurately program the higher-order motions of hollow 3D microstructures. For example, uniaxial/biaxial contraction and articulated folding (Figure 8) are among the motions that match well with the numerical predictions made by FEA. This convergence between simulation results and actual performance enhances our understanding of how these innovative micro/nanorobots behave during shape reconfiguration.

The time-dependent shape changing [108] belongs to the fourth dimension. However, the shape-mutation-induced functional changing brings the fifth-dimensional 5D ability of optical functions into the 4DM/NRs. The 4DM/NRs possess 5D biomimetic functions of sensory function [109,110], chameleon-like coloration [111,112,113,114], dynamic optical devices [115,116,117,118,119] with variable focus, information encoding/transmission [120], wettability tailoring [121], and even electronic skin [122]. The typical 5D optical devices can be represented by color-shifted photonic crystals [18] or smart biomimetic compound eyes [118]. Dynamic micro/nanoscale 4D optical devices inherently have better integration ability and tunable imaging mechanisms, leading to optical performance improvement.

For example, compound eyes have long been recognized for their potential in modern optics, but both natural and artificial compound eyes suffer from a limitation—they lack the ability to achieve variable-focus imaging due to their fixed focal lengths within individual ommatidia. The researchers of Ref. [118] sought inspiration from the tunable crystalline lens found in human eyes to overcome this limitation. They successfully fabricated smart stimuli-responsive compound eyes using bovine serum albumin (BSA) protein through the innovative technique of femtosecond laser fabrication (Figure 9). By leveraging the unique swelling and shrinking properties of BSA under different pH conditions, the research team created compound eyes with a tunable field of view ranging from 35° to 80°, as well as variable focal lengths within each ommatidium. This composite compound eye exhibited nearly 400% focal length tuning while maintaining a fixed tunable field of view.

To evaluate its imaging capabilities, the researchers of Ref. [118] conducted tests using the letter “F” as a test object. They captured images of the letter “F” using the inner, middle, and outer parts of the compound eye separately. They specifically examined the inner, middle, and outer ommatidia individually to assess their respective focusing results (Figure 9a–d). The quality of focus achieved by these ommatidia and the grayscale intensity distribution of focal spots is extracted from six symmetrical ommatidia belonging to each part (inner, middle, and outer). By studying these distributions, researchers gained insights into how well each part of the compound eye focused on specific objects.

In addition to optical applications, another potential trend of 4DM/NRs is the interdisciplinary application in bio-science [123,124,125,126,127,128,129,130,131,132,133,134,135,136,137]. To enable more intricate micro/nano cargo manipulation, such as encapsulation and release, in biological settings, it is crucial to equip microrobots with the ability to adapt and morph their shapes in dynamic environments (Figure 10). By combining magnetic propulsion with shape-morphing capabilities, Ref. [123] creates a remarkable shape-morphing micro crab capable of performing targeted microparticle delivery (Figure 10c–f). 

As a proof-of-concept demonstration, they have also designed a shape-morphing micro fish that can encapsulate a drug (doxorubicin (DOX)) by closing its mouth in phosphate-buffered saline with a pH of around 7.4. Moreover, 4DM/NRs successfully achieved localized HeLa cell treatment within an artificial vascular network by utilizing the “opening–closing” motion. Another example of bio-application via 4DM/NRs may be the robotic devices and the light-driven artificial organs demonstrated by researchers of Ref. [82], they developed a miniature aortic valve that could potentially revolutionize medical treatments and advance biomedical research. The fabricated robotic aortic valve is controlled by applied light to open or close (Figure 11), and the specific opening amplitude of the aortic valve under different light stimulation powers is tested. Potentially, these novel organs can be implanted into real human/animal organs as cardiac pacemakers or for other adjuvant therapy.

In recent years, an increasing number of bio-applications have emerged that make use of Four-Dimensional Micro/NanoRobots (4DM/NRs). These tiny robots hold tremendous potential for various biomedical applications. The continuous advancements in size optimization, motion control, and imaging technology have significantly contributed to unlocking the capabilities of these 4DM/NRs. Researchers are constantly working towards optimizing their size to ensure compatibility with biological systems. This allows for minimally invasive procedures and precise manipulation at the microscale. Moreover, improvements in motion control mechanisms enable these robots to navigate complex environments with accuracy and precision. They can be potentially guided through intricate pathways, such as blood vessels or tissues, facilitating targeted delivery and microcargo operations. Furthermore, the integration of imaging technology provides real-time monitoring and feedback during the operation of these 4DM/NRs. Overall, 4DM/NRs hold great promise as ideal platforms for complex micro cargo operations and on-demand drug release. Their ability to navigate within biological systems coupled with their precise manipulation capabilities makes them valuable tools in biomedical research and healthcare applications. Continued advancements in this field will further expand their potential impact on improving medical treatments and interventions.

## 6. Advantages of Current Miniaturized 4D Robots, Challenges, and Future

In the era of micro/nano 4D products, there simultaneously exist other competitive 4D products (Figure 12). The first one may be the microscale self-propelled bio-robots that use microorganisms [138] from the animal or plant world. As we know, there exist animal robots, such as trained dogs, bees, wasps, termites, or ants, being used as robots. On the micro/nanoscale, the available tiny biological robots can be microorganisms, cells [139], bacteria, eggs of insects, and even sperm. Peer researchers employ fish sperm cells as micromotors to interact with the surrounding environment to trigger a swimming speed, generating a snake-like moving path. The tiny microorganism moves and self-navigates to execute specific tasks. For example, aqua sperm [140], whose size is similar to that of bacteria, is reported to destroy biofilms that colonize medical and laboratory tubing. In addition, the other plant tissue robots are transformed by humans using modified cultivation engineering in the presence of functional nanoparticles, which happen through the cell growth media, and these particles are taken up into the plant tissue cells for remote magnet-driven motion ability.

The second competitor is two-aced colloidal Janus nanoparticles [141]. These anisotropic nanoparticles are synthesized by stepwise chemical methods, where one face of a Janus nanoparticle is colloidal (polymers, plastics, SiO_2_, and so on), and the other face is metal-deposited, such as iron, titanium, copper, or platinum for magnetic force or photocatalysis ability [142,143]. Janus nanoparticles generally show desirable dispersion characteristics in organic aqueous solutions, like surfactants, amd by rotating the magnetic field’s direction, the Janus particles move in the controlled direction. Another driving force is gas propulsion from chemical catalysis reactions to the photothermal effect [144]. The deposited metal decomposes organic solvent into gaseous substances under light projection to put themselves forward by the generated gas.

The third competitor is the modern integrated micro-robots using multi-material stepwise semiconductor fabrication [145], and digital control [146] for the swarm character, high intelligence [147], autonomy, and even wireless communication. The above-studied four main 4DM/NRs types are reshaping our modern bio-science, robotics, micro-electro-mechanical systems, and the Internet of Things by playing ever-increasingly important roles. However, their similarity, differences, advantages, and disadvantages vary a lot in terms of volumes, fabrication methods, application ranges, flexibility and self-recovery, and feature size, which can be found in the comparative table (Figure 13). In conclusion, laser synthesis is a green fabrication method [148] requiring no tedious semiconductor process, nor the cultivation of microorganisms, while maintaining bio-friendliness [149,150] and the high-order regulation of wanted functions.

## 7. Conclusions

Conclusively, the laser photochemically synthesized 4DM/NRs have compliant body structures made of smart materials, or their composites can be printed directly using the current digital computer-assisted laser system. The modern digitalized ultrafast laser system [148] promises high compatibility with compound materials and high complexity in structure/pattern design, allowing for the implantation of shape-reconfiguration memory during fabrication. It is worth affirming that the digital laser system provides high flexibility in nanostructured heterojunction or meta-mechanical structures to responsive materials for performance improvement.

The reviewed 4D laser-synthesized monolithic soft robots are capable of linear and turning locomotion at the micro/nanoscale for breaking the limitations of volume and controllability by traditional chemical/physical methods. By doping functional nanoparticles, the synthesized 4DM/NRs possess potential far beyond the currently reported robotic applications. Thereby, we strongly recommend the photochemistry method for customizing next-generation 4DM/NRs for envisioned applications. Using this method, the driving modes/programming strategies/spatial resolution and volume are versatile and applicable to various extreme conditions.

## Figures and Tables

**Figure 1 micromachines-14-01656-f001:**
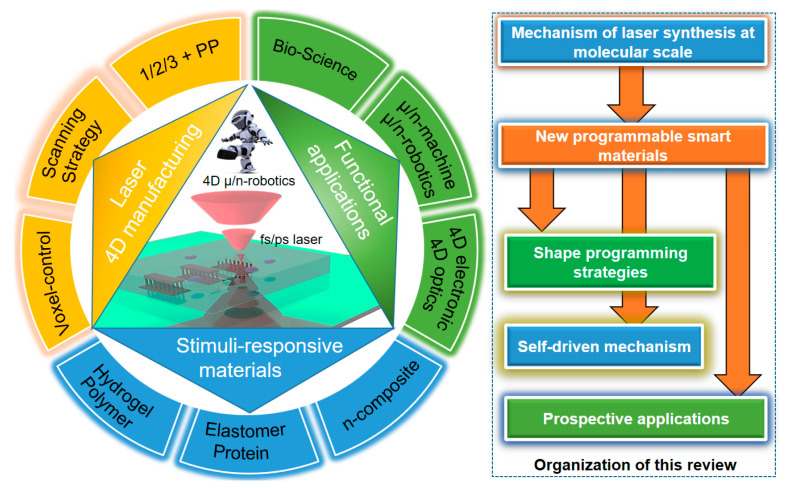
Schematic illustration of this review, covering three main sections: 1. laser-associated 4D manufacturing approaches and programming strategies around shape programming, which mostly operate on high-resolution two-photon polymerization and differ from macroscopic extrusion, material jetting, or other 3D printing methods; 2. stimuli-responsive materials, especially functional hydrogels/polymers, inorganic nonmetallic materials, and their nanocomposites; 3. future prospective interdisciplinary usage via 4D micro/nanofabrication is potentially applicable to micro/nanomechanics, adaptive optics, and bio-sciences. The organization logic throughout this manuscript follows this order: laser synthesis → smart materials → the programming strategies of 4D micro/nanorobotics → self-driven mechanism → application and comparison.

**Figure 2 micromachines-14-01656-f002:**
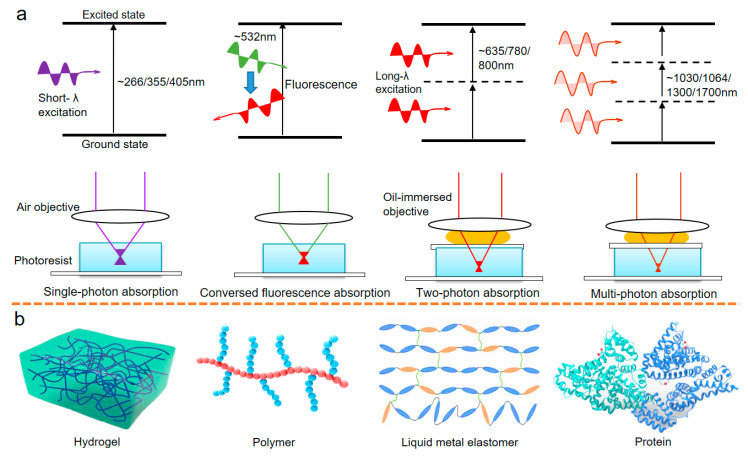
(**a**) Single–photon absorption at relatively short-wavelength laser projection, green-light-triggered fluorescence absorption, typical two-photon absorption, multi-photon absorption at infrared wavelength laser projection. (**b**) There are four representative smart materials used in laser photochemistry for robotic applications: stimuli-responsive hydrogel, shape memory polymer, liquid metal elastomer, and protein materials.

**Figure 3 micromachines-14-01656-f003:**
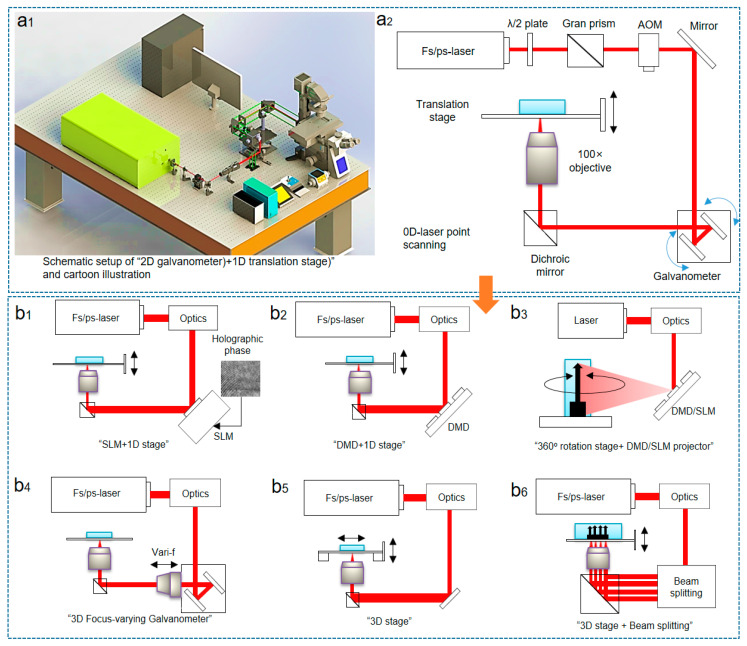
(**a_1_**) Illustration of laser direct writing system and (**a_2_**) its optical principle; (**b_1_**–**b_6_**) The current optical principle for high-freedom formation has developed to new categories: 2D spatial modulator (SLM) + 1D stage; the high-speed digital mirror (DMD) + 1D stage; the rotational substrate + projector; the newly-emerged 3D galvanometer with dynamically tunable focus lens; integrated 3D stage; 3D stage + beam splitting for paralleled fabrication.

**Figure 4 micromachines-14-01656-f004:**
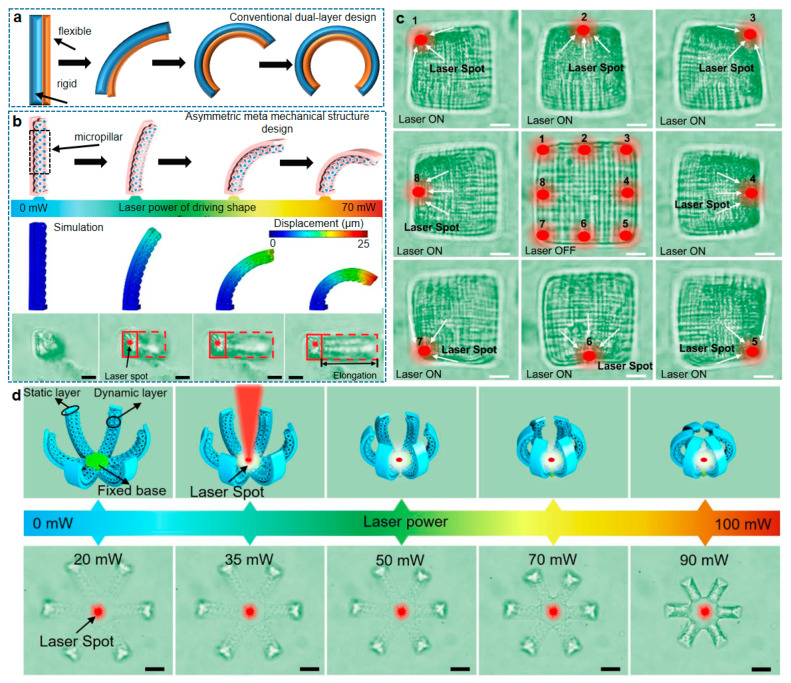
(**a**) The classical soft–rigid dual layer for shape bending. (**b**) The comprehensively enhanced shape reconfiguration of meta mechanical structure (biomimetic cilia). The simulation using finite element analysis matches well with experiment. (**c**) Optical micrographs of different shape transition states of woodpile photonic crystal microstructures regulated by the laser stimulation. There are eight positions of laser spot as indicated by the arrows. (**d**) Meta-structure-enhanced flower-shaped microclamp using a laser beam as photon stimuli. These figures are reproduced from Ref. [82] with copyright permission. The scale bar is 10 µm. The scale bars of (**b**–**d**) are 10 µm.

**Figure 5 micromachines-14-01656-f005:**
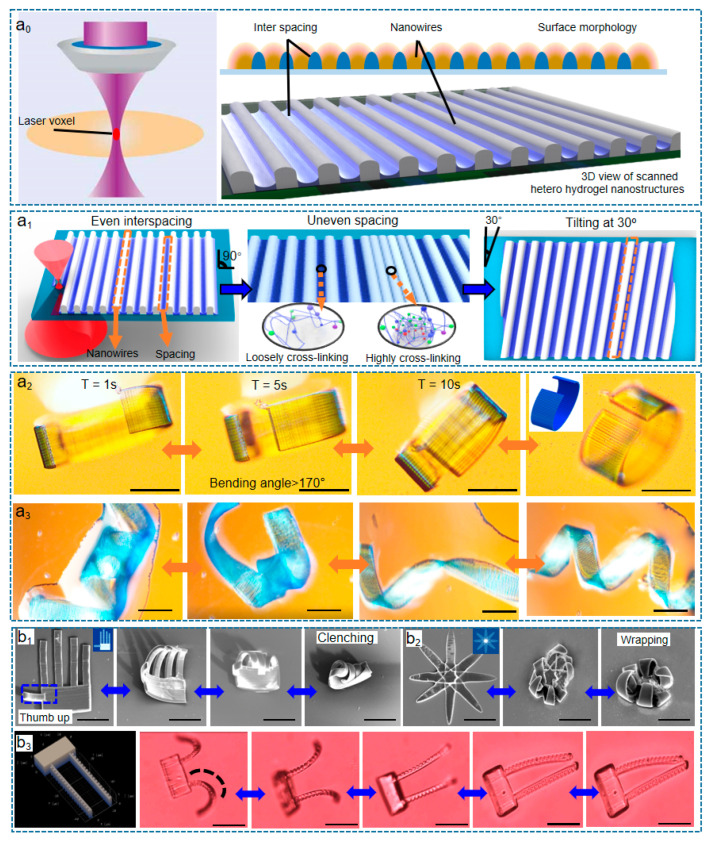
(**a_0_**) Illustration of laser focus voxel during the laser synthesis process. The red ellipse denotes the focused laser point inside the precursor material. The other figure is the section view of the laser-scanned interlaced nanowire/interspacing nanostructure reproduced from Ref. [18], which is known as a heterojunction nanostructure in Ref. [12]. (**a_1_**) The nanowire-based heterojunction scanning strategies in monolayer. (**a_2_**,**a_3_**) The accumulated intermolecular force realized “bracelet” and chiral torsion, respectively. (**b_1_**–**b_3_**) The out-of-plane bent biomimetic hand, flower, and grippers. These figures are reproduced from Ref. [12] with copyright permission from the authors. The scale bars of (**a_2_**,**a_3_**) are 50 µm, and the scale bars of (**b_1_**–**b_3_**) are all 50 µm.

**Figure 6 micromachines-14-01656-f006:**
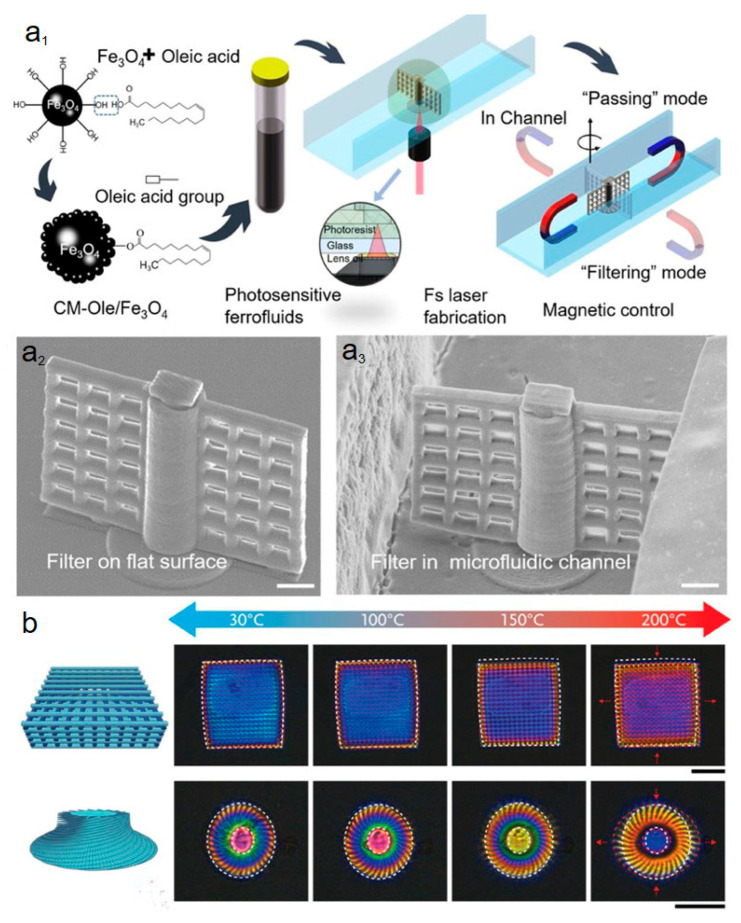
(**a_1_**) Fabrication procedures for the magnetically driven rotary microfilter reproduced from Ref. [86] with permission from © The Optical Society. (**a_2_**,**a_3_**) SEM images of the two-photon polymerization fabricated filter on a flat surface and embedded inside a fluidic channel. The scale bars are 20 μm. (**b**) Example of temperature response: a woodpile and spiral disk thermally change their dimension and lead to coloration change. Reproduced from Ref. [90]. The scale bars represent 20 μm.

**Figure 7 micromachines-14-01656-f007:**
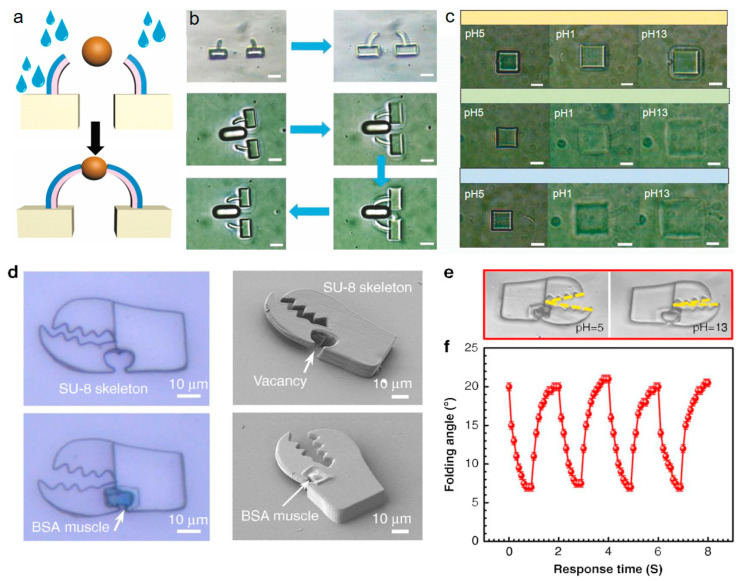
(**a**) Illustration of dual-layer structure shape changing by pH variation. Reproduced from Ref. [96]. (**b**) The reverse gripping action from a pair of designed protein cantilevers, where the scale bar is 5 μm. (**c**) The swelling/shrinkage of a cube shape made of proteins by two-photon polymerization. Reproduced from Ref. [96]. The scale bar is 10 μm. (**d**–**f**) The authors of Ref. [83] demonstrate a composite-material micro claw robot responsive to pH variation, where SU–8 photoresist works as the rigid part and BSA is bovine serum protein, which is polymerized on the claw to form the flexible muscle.

**Figure 8 micromachines-14-01656-f008:**
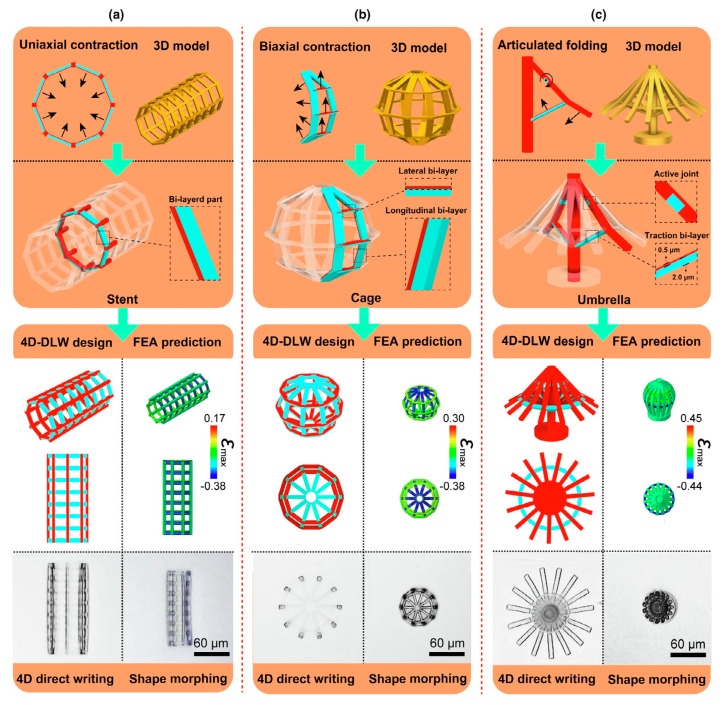
The authors of Ref. [104] present demonstrations of innovative technologies that enable the creation of complex 3D reconfigurable micro-architectures. These architectures are made possible through the use of advanced micromachines, such as microstents, microcages, and micro-umbrellas (**a**–**c**). These micromachines are capable of achieving various forms of reconfiguration, including rapid and precise uniaxial contraction, biaxial contraction, and articulated-lever folding. This level of flexibility allows for a wide range of applications in fields such as robotics and biomedical engineering. The process used to create these structures is known as 4DLW (four-dimensional direct laser writing). It involves several steps, including the embedding of deformation-amplifying mechanisms, the design of the 4DLW system, FEA prediction to optimize performance, the actual 4D direct writing process using lasers or other techniques, and finally, shape-morphing to achieve the desired configuration. The pictures are reproduced from Ref. [104].

**Figure 9 micromachines-14-01656-f009:**
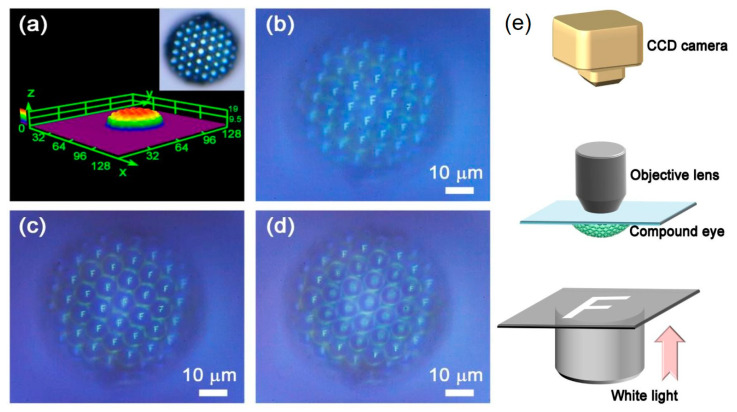
The focusing and imaging performances of a batch of protein-based compound eyes reproduced from Ref. [118] with the author’s permission. The compound eye under investigation is composed of ommatidia, each measuring 10 μm in diameter and 3 μm in height. (**a**) Displays a 3D laser confocal microscopy image capturing the detailed structure of the compound eye. To assess its imaging capabilities, the researchers analyzed the performance from the inner part to the outer part of the compound eye, as depicted in (**b**–**d**). (**e**) The schematic of the optical setup used to measure the imaging performance of compound eyes.

**Figure 10 micromachines-14-01656-f010:**
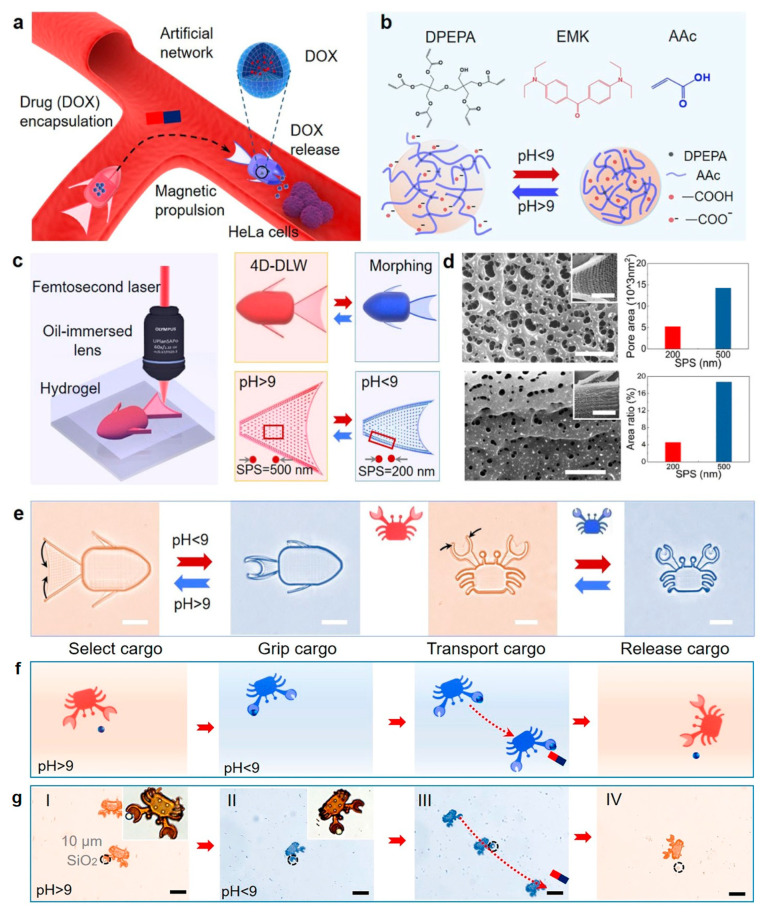
The researchers of Ref. [123] achieved one-step 4D printing of shape-morphing microrobots for HeLa cell treatment. The fabricated magnetic microrobots (in the shapes of crabs and fishes) can undergo shape-morphing to achieve targeted drug release for treating cancer cells. (**a**) Schematic of the magnetic shape-morphing microfish (SMMF) for targeted doxorubicin (DOX) release to treat cancer cells by shape-morphing. (**b**) Main compositions and schematic of pH-responsive hydrogels, where AAc is acrylic acid, DPEPA is the cross-linker thdipentaerythritol penta acrylate, and EMK is the photoinitiator 4,4′–bis–(diethylamino)benzophenone. (**c**) Four-dimensional printing with a designable point density to encode shape-morphing. (**d**) Scanning electron microscopy (SEM) images of the gel pores in the tail and body (inset images). (**e**) Optical images of the 4D fish and crab, illustrating the opening and closure of the fins and claws, respectively. (**f**,**g**) Schematic procedures and time-lapse images of I. selecting, II. tightly gripping, III. transporting, and IV. releasing targeted cargo by the 4D crab. Scale bars: (**d**) 1 μm, and inset images, 5 μm; (**e**) 25 μm; (**f**,**g**) 50 μm. All pictures here are reproduced from Ref. [123].

**Figure 11 micromachines-14-01656-f011:**
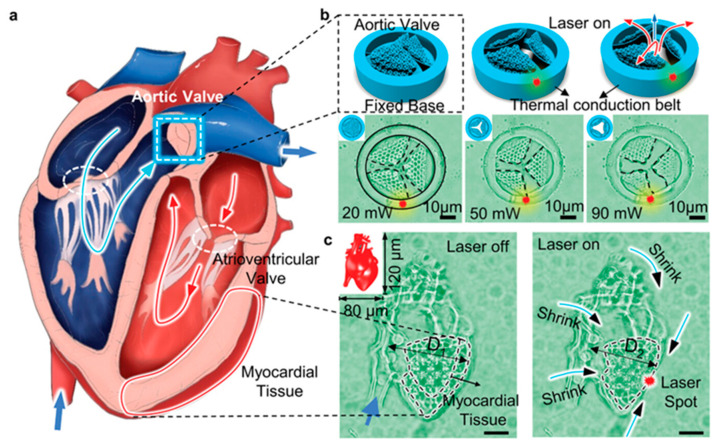
The researchers of Ref. [82] have successfully printed and demonstrated a functioning microscale artificial heart. Their schematic of the heart structure is depicted in (**a**). To assess its functionality, the opening amplitude of the aortic valve is examined under different light stimulation powers, as shown in (**b**), and optical micrographs captured the changes in valve opening as a result of varying light stimulation. Furthermore, optical micrographs were taken to compare the appearance of the microheart with and without light stimulation, as seen in (**c**). It is worth noting that this remarkable piece of technology has incredibly small dimensions, measuring merely 80 × 120 × 60 µm^3^, as indicated by the schematic illustration inset, in which the scale bar denotes 20 µm.

**Figure 12 micromachines-14-01656-f012:**
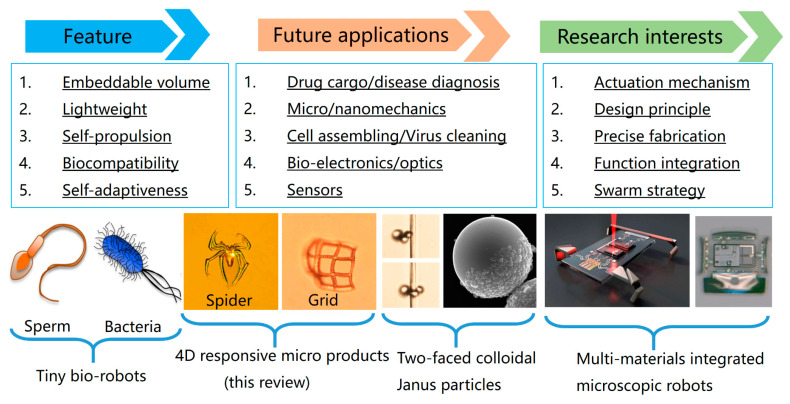
The other existing 4DM/NR types reported, including their features, prospective applications, and research interests.

**Figure 13 micromachines-14-01656-f013:**
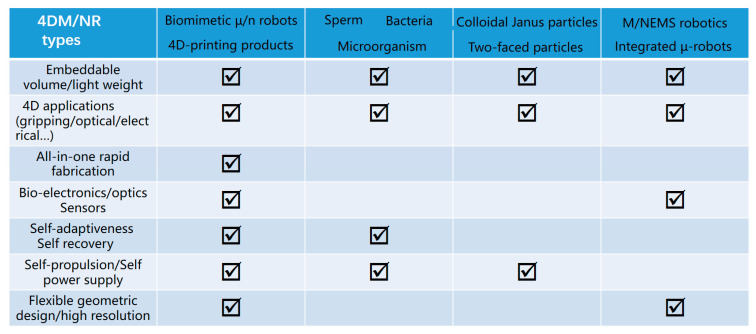
Comparative analysis of four different currently studied 4DM/NRs in terms of applications, fabrication, unique advantages, design capability, and feature size, where laser photochemically synthesized 4D robots possess the most comprehensive advantage, slowing the application range covering micro/nano optics, electronics, and robotics with flexible geometric design outperforming the human-cultured microorganisms or nanoparticle-shaped Janus robotics. Moreover, the laser synthesis allows multi-function integration in tiny volumes requires only one-step fabrication, which is much more efficient and beneficial to shorten the design circles.

## Data Availability

Data is contained within this article.

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
