# Peer review of "Four-Dimensional Micro/Nanorobots via Laser Photochemical Synthesis towards the Molecular Scale"

_micromachines, 2023, doi:10.3390/mi14091656_

Round 1
Reviewer 1 Report
The presented paper gives a brief overview about the emerging field of 4D robotics, fabricated with laser photochemical synthesis.
The topic is undoubdtedly interesting and important to discuss. The paper cites important previous research work from prestigious journals. The overall structure meets the criteria of scientific writing, however, I recommend some major changes in order to considarate the publication in Micromachines:
- It is not clear, how the literature structure has been carried out (e.g. what were the search terms / mesh, which databases have been used, how many articles have been selected originally, and what were the exclusion and inclusion criteria).
- In the conclusion, I miss the demonstration of the practical use cases of 4D micro scale robotics. What are the main possible uses cases in biological systems and healthcare?
- In the introduction part, it would be recommended to give a general explanation to the defition of "4D structures" (why do we call it 4D), also, 4D printing should be briefly discussed (including the connection with "tradtitional" 3D printing)
In conclusion, the paper has a merit, but the mentioned information should be included before considering it for publcation.
Respectfully yours,
The overall quality is good, but professional language editing is strongly recommended, and the in some cases the pharsing and wording sounds too general.
Author Response
Dear Reviewer1,
We would like firstly to express our sincere gratitude for your time during the reviewing process. Your feedback, questions, and comments are helpful to refine this manuscript further. We appreciate your deep consideration and expertise. Your opinions will be carefully discussed, and your questions will be answered one by one.
In the resubmitted version of the manuscript, we have made changes and emphasized certain sections, which are highlighted in red for your convenience during the checking process.
Once again, we thank you for your valuable guidance.
Reviewer1:
Open Review
Your concerns have prompted us to conduct a thorough investigation. As authors, we are open to any additional suggestions or comments that you or the reviewer may offer in the future. We are willing to make any necessary alterations to fully address or misinterpret the answers.
During the review process, we have taken extra care in selecting appropriate language based on current literature and have thoroughly checked for spelling, punctuation, and grammar errors to ensure clear descriptions. Any identified mistakes or areas for improvement have been addressed accordingly.
Comments and Suggestions for Authors
The presented paper gives a brief overview about the emerging field of 4D robotics, fabricated with laser photochemical synthesis.
Answer: We are so lucky to have your time and consideration. We feel deeply understood by our excellent peer researchers, and we thank you for all the careful evaluation and constructive comments.
The topic is undoubdtedly interesting and important to discuss. The paper cites important previous research work from prestigious journals. The overall structure meets the criteria of scientific writing, however, I recommend some major changes in order to considarate the publication in Micromachines:
Answer: Your positive acknowledgment encourages us a lot, and we understand the necessity of major changes. A corresponding systematic modification around the fundamental concept, paragraph organization, presentation logic, and analysis is conducted in reviewing process. Your deep insight and suggestion will help us improve this review in multi-fold fields.
- It is not clear, how the literature structure has been carried out (e.g. what were the search terms / mesh, which databases have been used, how many articles have been selected originally, and what were the exclusion and inclusion criteria).
Answer: Thanks for your pointing out the vague place, we apologize for the unclear definition of the search terms, database, and related original articles. As you pointed out, we indeed have not clearly explained the exclusion and inclusion criteria, this review lacks a clear conceptual boundary of the discussed 4D micro-nanoscale robotics.
As a correction, the keywords are rewritten to confine the specific goal as the search terms, which center on the micro nanorobots and related laser preparation. The organization logic throughout this manuscript follows this order: laser synthesis à smart materials à the programming strategies of 4D micro/nanoroboticsà self-driven mechanism à application and comparison. Therefore, figure 1 is re-plotted with an added framework of organization as below.
To avoid losing the track of motif in reading, this re-submitted version only focuses on small-volume biomimetic robotics using laser synthesis methods. For clarity, we rewrote the Abstract part and the beginning paragraphs to limit the review scope, only micro/nano laser structured robotics are our review object, and all current robotics at macroscopic volume (larger than 1 millimeter) are excluded. The databases come from the reports of mainstream scientific journals (including Wiley, MDPI, and Springer).
We will maintain the comparison part about the other small robotics, although their fabrications deviate from the reviewed laser-induced synthesis method, their volume still distributes at the same or similar scale to the central micro/nanorobotics.
- In the conclusion, I miss the demonstration of the practical use cases of 4D micro scale robotics. What are the main possible uses cases in biological systems and healthcare?
Answer: This is a great question. We understand your concern, the practical use/application is truly an important part that deserves highlighting.
Have to say, the bio-robots made by laser synthesis denote a frontier field in additive manufacturing technique, very few reports successfully demonstrate the comprehensive applications in bio-science by laser synthesis, and the specific applications in treating cancers/tumors, therapeutics have not been realized, therefore, this may disappoint the reviewers and readers.
The main possible usage in biological systems and healthcare can be represented by drug delivery robotics, like the discussed Figure 10, the laser-synthesized micro/nano fishes/crabs are non-contact driven to locate the cancer cells and deliver medicine. To highlight the usage in biological systems, the description of Figure 10 is rewritten in more detail. Figure 10 only verifies the potential of using micro/nano 4D robots in drug delivery at an early stage.
We agree to reintroduce the demonstration about usage in the biological system and healthcare. Therefore, another new application of laser-synthesized micro/nano 4D robots for simulating heart beating is added as below.
The newly-added microscale artificial heart as Figure 11, a) Schematic of a heart structure. b) Illustration and optical micrograph of the opening amplitude of the aortic valve under different light stimulation powers. c) The optical micrographs of the micro heart without and with light stimulation. The illustration inset shows a schematic of a micro heart with a dimension of merely 80 × 120 × 60 µm3. The scale bar is 20 µm. The added Figure 11 may su
Besides, the other micro/nano 4D robots not fabricated by laser synthesis have been reported to implement the mission of single-cell surgery (ref.129), treating tumors (ref.129), DNA Vaccine Delivery(ref.132), cancer therapeutics(ref.136), targeted virotherapy of homologous carcinoma (ref.139), and so on.
- In the introduction part, it would be recommended to give a general explanation to the defition of "4D structures" (why do we call it 4D), also, 4D printing should be briefly discussed (including the connection with "tradtitional" 3D printing)
Answer: We would like to accept your constructive recommendation, the explanation about "4D structures" is the chief concept covering the entire review. We explain about 4D structrues and 4D printing as follolowing: “ there has been a growing interest in the development of dynamic shape-transforming 3D printed structures, which is considered the next big breakthrough in additive manufacturing techniques. This innovative research aims to create 3D printed structures that can actively change their shape over time. The term “4D printing” was first introduced by Tibbets et al. in 2013 to describe this concept of creating objects with the ability to transform their shape dynamically. Since then, researchers and engineers have been exploring various materials and design strategies to realize the potential of 4D printing and its applications across different industries. This exciting field holds promise for advancements in areas such as robotics, healthcare, architecture, and more.” at the beginning of this review now.
In conclusion, the paper has a merit, but the mentioned information should be included before considering it for publcation.
Respectfully yours,
Answer: We are grateful for your kind instruction, professional suggestion, and positive recommendation for publication. All your concerns will be
Comments on the Quality of English Language
The overall quality is good, but professional language editing is strongly recommended, and the in some cases the pharsing and wording sounds too general.
Answer: We feel excited to get your praise, and we sincerely apologize for our plain and ordinary language. In the short-time reviewing step, we use language tools like Grammarly, 1Checker, and some artificial intelligence-based paraphrase boxes to optimize the phrase and wording.
Reviewer 2 Report
This review introduced the recent development of micro/nano 4D- 24 freedom laser photochemical manufacturing incorporating active materials and shape-program- 25 ming strategies. The comparison with other chemical/physical fabricated micro/nanorobots further 26 explained the advantages of laser-synthesized micro/nanorobots. The manuscript is very standardized, in line with the theme of the journal, and is highly recommended. But the following issues need to be addressed.
1.How do these material structures in Figure 5 contribute to or promote the bionic hand, flower and gripper?
2.Is there any difference between a1, a2 and a3 non-contact magnetic field driving in Figure 6?
Please check the full text in English for case of case inconsistency
Author Response
Dear Reviewer2,
We would like to express our sincerest apologies for any negative impressions caused by the errors and inappropriate descriptions in our manuscript. In line with your recommendations, we have taken them into careful consideration and made significant modifications throughout the entire manuscript. We aim to reduce any potential misunderstandings caused by literal errors and ensure scientific clarity. We have diligently verified and reorganized the content as per your suggestions. To facilitate the checking process, we have highlighted these modifications in red. Additionally, we have also included an edited version of the same content without any color markings for editing.
In summary, we are committed to rectifying language errors and enhancing the clarity of our discussions and evidence. We appreciate your patience and guidance throughout this process.
Reviewer2:
(x) I would not like to sign my review report
( ) I would like to sign my review report
Quality of English Language
( ) I am not qualified to assess the quality of English in this paper
( ) English very difficult to understand/incomprehensible
( ) Extensive editing of English language required
( ) Moderate editing of English language required
(x) Minor editing of English language required
( ) English language fine. No issues detected
Comments and Suggestions for Authors
This review introduced the recent development of micro/nano 4D-freedom laser photochemical manufacturing incorporating active materials and shape-program-ming strategies. The comparison with other chemical/physical fabricated micro/nanorobots further explained the advantages of laser-synthesized micro/nanorobots. The manuscript is very standardized, in line with the theme of the journal, and is highly recommended. But the following issues need to be addressed.
1.How do these material structures in Figure 5 contribute to or promote the bionic hand, flower and gripper?
Answer: Thank you for asking this question. The material used in Figure 5 is a hydrogel and scanned by laser to form monolayer heterojunction nanostructures, the hydrogel materials cross-linked in a monolayer interlaced structure, leading to an ultrathin heterojunction-like structure.
Correspondingly, the revised manuscript has a new Figure 5, with added Figure 5a0, the material structure is an ultrathin heterojunction-like structure. The mechanism to promote the hand, and flower is thermally shrinkage, while the gripper is realized by the solvent stimuli-induced non-covalent bonding. We add a brief introduction to explain the non-covalent bonding effect:
“The shape reconfiguration of 4DM/NRs can also be attributed to the non-covalent bonding effect. Non-covalent bonding refers to the interactions between molecules or particles that do not involve the sharing or transfer of electrons. These interactions include hydrogen bonding, van der Waals forces, electrostatic interactions, and hydrophobic interactions.
The non-covalent bonding effect plays a crucial role in the structural stability and functionality of 4DM/NRs. By harnessing these intermolecular forces, researchers are able to design and fabricate intricate structures with precise control over their shape transformation. The non-covalent bonds act as reversible connections, allowing robots to undergo shape reconfiguration in response to external stimuli.”
2.Is there any difference between a1, a2 and a3 non-contact magnetic field driving in Figure 6?
Answer: Thanks for asking this question.
The three figures all describe the same magnetic force. But Figure 6a1 is different from 6a2 and 6a3. Actually, Figure 6a1 describes the fabrication procedure, where the laser-fabricated filter needs post-process to capture magnetic nanoparticles. The magnetic particles contribute to the non-contact magnetic force-driven filter rotating in the liquid state.
Figures 6a2 and 6a3 are filters on the plane and inside a microfluidic channel, their structures are the same, but application situations are different. In the revised version, a brief explanation is added to the surrounding paragraphs, and the caption of Figure 6 is slightly corrected to add the scale bar 6a2 and 6a3.
Comments on the Quality of English Language
Please check the full text in English for case of case inconsistency
Answer: A check covering the entire manuscript is conducted, and the changed words/sentences are marked in red color for your review.

Reviewer 3 Report
The paper of Tao et al., entitled “4D-freedom micro/nanorobots via laser photochemical synthesis towards the molecular scale” is presenting a detailed literature review on the fabrication and functionality of miniaturized micro/nanorobots obtained by laser photochemistry. The paper is well structured, comprehensive, most of the cited literatures is recent and very recent (last 3-4 years) and published in high-ranked journals, so I consider it could be useful for the readers of Micromachines journal. However, there are some different parts of the manuscript that could be improved in order to increase the presentation quality of this paper, before publishing. My suggestions are listed in the following:
Refs 50 and 54 are the same. Please correct!
Figure 4.b is not sufficiently explained in the text. How laser spot influence the behavior of the meta structure? Scale-bar value is missing in the figure. In the caption of din figure is cited ref [78], while in the text is mentioned [82]. Figure 4.c is not discussed at all in the text of the manuscript!
Lines 246-247 – “laser scanning maintains single voxel resolution” – which is the dimension of the voxel?
Lines 253-254 – “High-order directional shape reconfiguration by manipulating nanowires.” – this phrase should be revised. The dimension of the nanowires and of the laser focused should be introduced.
More details about Figures 5 should be given in the manuscript. Scale-bars are missing. Fig 5.c mentioned at line 250 is missing.
More details about Figures 6 should be given in the manuscript. Scale-bars are missing. Only brief description in the captions of the figures is not sufficiently for the readers.
Lines 287-288 – “The second driven force is the intermolecular effect-induced dynamic force by responsive stimuli of various solvents [18, 90, 91].” – but the references [90-91] refers to temperature-responsive structures. Please correct!
Figure 7 is not cited in the text and should be better explained. Scale-bars are missing in Fig 7.b.
Line 356 – “where 4DLW denotes the four-dimensional direct writing technique, and FEA denotes finite element analysis” – the acronyms were already introduced before.
More details about Figures 8 should be given in the manuscript. Only brief description in the captions of the figures is not sufficiently for the readers.
Lines 367-368 – “The focusing and imaging performances of the eyes are much magnified (Figure 9).” – Unclear! Brief details about fabrication of biomimetic compound eye and the setup in Fig 9 “(e)” (right) should be introduced. BSA – is the acronym for Bovine Serum Albumin.
Caption of Fig 10 – SMMFs acronym should be explained.
Line 429 – I suggest to replace “minimized” with “miniaturized”. This could be applied in the abstract as well.
Line 473 – “Although there are several challenges in developing untethered soft robots including ease of fabrication, efficient actuation system, and multiple integrated functionalities.” – Unclear!
Line 475 – “This study proposes a fully 3D printed, monolithic soft robot capable of linear and turning locomotion.” – this paper is a review, which study?!
Figure 12 and the comparison are better fitted at the end of section “6. Advantages among current minimized 4D robots, challenges, and future” in my opinion.
Some phrases are unclear and should be revised!
Author Response
Dear Reviewer3:
We feel grateful to your careful checking and kind instruction. We would like to express our sincere gratitude for your feedback throughout this manuscript. Your careful checking and valuable questions are much appreciated. We know that your concern is important to correct our errors and improve presentation quality, and we accept all your comments. In the revised version, we try to fully revise this manuscript according to the questions one by one.
We have made it a priority to highlight and explain the changes by highlighting them in red for your review. Anyway, thank you once again for your time and expertise.
Reviewer3:
Open Review
(x) I would not like to sign my review report
( ) I would like to sign my review report
Quality of English Language
( ) I am not qualified to assess the quality of English in this paper
( ) English very difficult to understand/incomprehensible
( ) Extensive editing of English language required
(x) Moderate editing of English language required
( ) Minor editing of English language required
( ) English language fine. No issues detected
Comments and Suggestions for Authors
The paper of Tao et al., entitled “4D-freedom micro/nanorobots via laser photochemical synthesis towards the molecular scale” is presenting a detailed literature review on the fabrication and functionality of miniaturized micro/nanorobots obtained by laser photochemistry.
Answer: We sincerely welcome all your criticism and constructive suggestion, and thanks for your in-depth understanding of the reviewed topic. All your comments will help us to improve this manuscript for a better quality and scientific meaning.
The paper is well structured, comprehensive, most of the cited literatures is recent and very recent (last 3-4 years) and published in high-ranked journals, so I consider it could be useful for the readers of Micromachines journal.
Answer: We are encouraged by your praise of “well structured, …very recent”, your careful reading and understanding of our poor work is much appreciated, and we sincerely thank you for your acknowledgment and positive recommendation.
However, there are some different parts of the manuscript that could be improved in order to increase the presentation quality of this paper, before publishing. My suggestions are listed in the following:
Answer: We, all authors, are open to your questions or inquiries. We believe your suggestion is highly valuable for improving scientific quality, and we will follow your suggestion to modify the revised version until all issues be fully addressed.
Refs 50 and 54 are the same. Please correct!
Answer: Thanks for pointing out this duplication, we felt embarrassed by this error. Ref.50 is” 50. Liu, S.; Dong, X.; Wang, Y.; Xiong, J.; Guo, R.; Xiao, J.; Sun, C.; Zhai, F.; Wang, X. 4D Printing Of Shape Memory Epoxy For Adaptive Dynamic Components. Adv.Mater.Technol. 2023, doi:10.1002/admt.202202004.”, and Ref. 54 is referred to another article “Zhao,W.; Li,N.N.; Liu, L.W.; Leng, J.; Liu, Y.J. Mechanical behaviors and applications of shape memory polymer and its composites. Appl. Phys. Rev. 2023, 10, 011306.” now. The new Ref. 54 supports the shape-morphing mechanism of shape memory polymer in the cited sentence.
Figure 4.b is not sufficiently explained in the text. How laser spot influence the behavior of the meta structure? Scale-bar value is missing in the figure. In the caption of din figure is cited ref [78], while in the text is mentioned [82]. Figure 4.c is not discussed at all in the text of the manuscript!
Answer: To better explain the shape bending, The scale bar of Figure 4b is 10 μm.
The re-produced Figure 4b is added to a group of numerical simulations about the meta-mechanical micropillar-enhanced cilia, where bending elongation of the micropillar cilia gradually increases as the increase of light stimulation. Figure 4c is newly-added to show the photon responsiveness on a woodpile photonic crystal, where the laser projection generates a thermal effect, and re-shape the woodpile shape according to the laser position. The Figure 4d reflected the detailed relationship between laser power and re-configured shape.
As the laser power reached 70 mW, the cilia achieved a maximum elongation of 26 µm due to the saturation of the water desorption characteristics of the CNNC hydrogels even if the power continued to increase. We have also conducted a finite element simulation of the laser-induced bending effect . The inset of Figure 4b shows the simulated bending geometry of a micropillar cilia with light stimulation, which exhibits an excellent consistency with the experimental results.
Figures 4.c is changed to Figure 4.d with more details, and discussed with the newly-added 4.c as below:
” Recently, these fundamental shape-programming strategies evolve by introducing meta-mechanical structures into the multi-layers, for instance, our peer researchers build small 3D micro cage-like micropillar meta structures [82] to replace the planar soft layer along with the rigid layer, creating unique meta-mechanical micropillar-triggered shape configuration of a cilia using two-photon polymerization (Figure 4b). These me-ta-mechanical micropillar cells present anisotropic shape morphing and strong adher-ence to rigid layers, ensuring high-order direction in reverse photothermal shape recon-figuration, and a spatial resolution of shape programming depends on the volume of a single micropillar at the microscale.
In the work of [82], one remarkable advantage of smart hydrogel is its ability to un-dergo precise and programmable light-induced transformations with exceptional spatial resolution. In an experiment, researchers focused a laser beam (approximately 1 µm²) on either the face or edge of a cubic woodpile structure measuring 50 × 50 × 50 µm³ (length × width × height, Figure 4c). Upon exposure to the laser, the structure exhibited a fascinating response - it shrank around the laser focus area and rapidly assumed various stable 3D morphologies. This dynamic behavior demonstrates the versatility and controllability of smart hydrogels in achieving complex shape transformations at microscopic scales. As seen in Figure 4d, researchers present a 3D micro clamp device consisting of a circular ar-ray of six micropillars. The device’s functionality is demonstrated through optical micro-graphs taken under different light stimulation powers in an aqueous environment.”.
Lines 246-247 – “laser scanning maintains single voxel resolution” – which is the dimension of the voxel?
Answer: The laser voxel presents Gauss distribution in optical engineering, the laser is focused by the objective lens in the microscope, and again focused inside the photoresist (the smart materials discussed in this review), the focused laser voxel is an eclipse.
Accordingly, a new inset figure, Figure 5a0, is inserted into figure 5, around former Lines246-247, the caption description is “(a0) the illustration of laser focus voxel during the laser synthesis process, the red ellipse denotes the focused laser point inside precursor material. The other figure is the section view of the laser-scanned interlaced nanowire/interspacing nanostructure, or known as heterojunction nanostructure;”
The added Figure 5a0
As illustrated, the volume of the laser focus (known as “voxel”) directly affects the scanned nanowires, the photon polymerization reaction happens only inside the “voxel”. The typical voxel shape focused by a Gauss laser beam is an eclipse at the nanoscale, therefore, during scanning, the volume of formed nanowires (its height, linewidth) depends on the laser voxel. As a result, the spatial resolution of the laser scanning-induced nanowires, patterns, and structures is determined by the specific volume of the laser voxel.
Lines 253-254 – “High-order directional shape reconfiguration by manipulating nanowires.” – this phrase should be revised. The dimension of the nanowires and of the laser focused should be introduced.
Answer: Thanks for asking this question!
We apologize for our haste in writing. For clarity, this sentence is rewritten into “Direction of shape bending/expansion is perpendicular to the laser-scanned nanowires, therefore, the bending direction can be controlled by adjusting the laser scanning direction.” The dimension of the laser-scanned nanowires depends on the laser focus directly, when the voxel of laser focus is small, the linewidth and height of nanowires are confined. To further demonstrate the nanowires-consisted nanostructure, another section view of the nanowire-consisted structures is inserted into Figure 5a0.
The added section view is a laser-scanned nanowire/interspacing nanostructure, or known as heterojunction nanostructure in our previously-reported work. This figure is combined with Figure 5a1, and 5a2-5b3 to illustrate the nanowire
More details about Figures 5 should be given in the manuscript. Scale-bars are missing. Fig 5.c mentioned at line 250 is missing.
Answer: As you indicated, the scale bars of Figures 5a1-5b3 are added into the caption. And detailed word experession about figure 5 are given around this figure.
Discussion about Figures 5a is as following:
“ The direction of shape bending/expansion is perpendicular to the laser-scanned nan-owires, therefore, the bending direction can be controlled by adjusting the laser scanning direction. The interactive materials used in Ref. [18] is polyethylene incorporated N-isopropylacrylamide, we identify chiral torsion, anisotropic deformation, and site-specific mutation, where the hydrogel structures could be selectively modified or al-tered at specific locations. The hydrogel nanowires demonstrated spontaneous self-repairing capabilities, allowing them to recover their original shape or functionality after being damaged or deformed. This reusability makes them promising candidates for micro/nano robotics. Beneficially, the dimension of the 1D nanowire is traceable to the 0D voxel (Figure 5a) of laser focus [81], therefore, monolayer or formed 2D metastructures re-alize nano-accuracy programming based on the spacing and linewidth of the nanowire, without losing shape reconfigurable capacity, which promises future molecule-level ro-bots.”
More details about Figures 6 should be given in the manuscript. Scale-bars are missing. Only brief description in the captions of the figures is not sufficiently for the readers.
Answer: Thanks for your requirement of detailed information. The background knowledge and introduction about Figure 6 are given now.
“This innovative microfilter enables the switching between two distinct modes: filter-ing and passing. One notable advantage of this magnetically driven rotary microfilter is its ability to perform multi-mode filtering functions. For example, it can effectively capture particles with a size of 8 µm while allowing particles measuring 2.5 µm to pass through. Furthermore, it can also accommodate situations where both types of particles need to be passed through without filtration. The responsive characteristic of the microfilter, enabled by magnetic control, significantly enhances its reusability. This means that the microchip can be utilized repeatedly for various applications without compromising its performance or filtering capabilities.
As you indicated, we introduced the meaning of FEA in both of text and figure caption, this redundant explanation is unnecessary.”
And the missed scale bar is in caption now.
Lines 287-288 – “The second driven force is the intermolecular effect-induced dynamic force by responsive stimuli of various solvents [18, 90, 91].” – but the references [90-91] refers to temperature-responsive structures. Please correct!
Answer: Thanks for your finding this error. It is inappropriate to cite Refs.[90,91] to support the viewpoint here. As you pointed, Refs. [90, 91] has been cited in the previous paragraph to support the temperature-induced 4D shape mutation, and Ref. [90] is reproduced in Figure 6. We are sorry for this erroneous citation here, it has been deleted into “…various solvents [18].”
Figure 7 is not cited in the text and should be better explained. Scale-bars are missing in Fig 7.b.
Answer: We sincerely thank you for this kind indication, your careful and kind pointing of our errors are sincerely appreciated. We totally accept your opinion. The citation sentence is “pH changes induce structural deformation (as demonstrated in Figure 7) …”, and description is:
“Figure 7 presents a novel approach to fabricating micromechanical devices by utilizing pH-responsive materials with nonuniform internal lattice density. The technique employed in this research is the femtosecond laser two-photon polymerization, which allows for the flexible design and high-precision production of 3D micro-nano devices. The increase in scanning step length from 100 nm to 200 nm led to a higher swelling ratio of BSA micro-nano blocks [96] . This can be attributed to a decrease in the cross-linking density of the processed structures. In a pH 1 solution, the swelling ratio was approximately 1.9 when the step length was 200 nm. On the other hand, in a pH 13 solution, the swelling ratio reached 265. When using a step length of 100 nm in an alkaline solution, the swelling ratio was found to be 76.7% of that observed in structures fabricated with a step length of 200 nm..”
Line 356 – “where 4DLW denotes the four-dimensional direct writing technique, and FEA denotes finite element analysis” – the acronyms were already introduced before.
Answer: As you indicated, we introduced the meaning of FEA in both of text and figure caption, this redundant explanation is unnecessary.
More details about Figures 8 should be given in the manuscript. Only brief description in the captions of the figures is not sufficiently for the readers.
Answer: Yes, we accept this suggestion, and a detailed introduction about Figure 8 are given now:
“Versatile shape-programming strategies and driven mechanisms give the 4DM/NRs huge capability and ability for broadband applications. The fundamental shape-reconfigurable application of 4DM/NRs is representative mechanical microactua-tors of expansion, bending, motion, and gripping. To enhance the control over shape transformations while minimizing response time, researchers have been exploring nanofabrication approaches with sophisticated designs. A groundbreaking four-dimensional microprinting strategy is introduced for constructing 3D-to-3D shape-morphing micromachines [104] in a single-material-single-step mode. By utilizing direct laser writing, the researchers are able to spatially distribute heterogeneous stimu-lus-responsive hydrogels into arbitrary 3D shapes with sub-micrometer features. The ma-terial properties of the hydrogels, including crosslinking densities, stiffnesses, and swell-ing/shrinking degrees, can be precisely modulated by programming the exposure dosage of femtosecond laser pulses.”
Lines 367-368 – “The focusing and imaging performances of the eyes are much magnified (Figure 9).” – Unclear! Brief details about fabrication of biomimetic compound eye and the setup in Fig 9 “(e)” (right) should be introduced. BSA – is the acronym for Bovine Serum Albumin.
Answer: As pointed out, Figure 9 is reproduced for simplicity and clarity. Another paragraph introducing the work of Ref. [118] is added as below:
“ For example, compound eyes have long been recognized for their potential in mod-ern optics, but both natural and artificial compound eyes suffer from a limitation - they lack the ability to achieve variable-focus imaging due to their fixed focal lengths within individual ommatidia. Researchers of Ref. [118] seek inspiration from the tunable crystal-line lens found in human eyes to overcome this limitation. They successfully fabricated smart stimuli-responsive compound eyes using bovine serum albumin (BSA) protein through the innovative technique of femtosecond laser fabrication (Figure 9). By leveraging the unique swelling and shrinking properties of BSA under different pH conditions, the research team create compound eyes with a tunable field of view ranging from 35° to 80°, as well as variable focal lengths within each ommatidium. This composite compound eye exhibited nearly 400% focal length tuning while maintaining a fixed tunable field of view.”
Figure 9 is reproduced into a simplified version as below:
The capation and description of figure 9 are correspondingly changed for background information of compound eyes and its functional test.
Caption of Fig 10 – SMMFs acronym should be explained.
Answer: We are sorry for the confusion by the missed explanation, SMMF denotes the “shape-morphing microfish” fabricated by laser fabrication. We added the specific meaning of “shape-morphing microfish” into the caption of Fig 10, and put “SMMF” into a pair of parentheses, (SMMF), as an acronym. The other sentences are slightly rewritten as well.
Line 429 – I suggest to replace “minimized” with “miniaturized”. This could be applied in the abstract as well.
Answer: We believe it is a good idea, and replace minimized” with “miniaturized”. “miniaturized” is more specific and professional to describe the small volume object, especially to the micro/nanoscale (known as molecular scale). The replacement is done in the Abstract part and title of Section 6 as suggested.
Line 473 – “Although there are several challenges in developing untethered soft robots including ease of fabrication, efficient actuation system, and multiple integrated functionalities.” – Unclear!
Answer: Thanks for pointing out this redundant sentence. We consider Line 473 and Line 475 as complete expressions, and we agree the subordinate clause is verbose, thus, we directly delete this subordinate clause, and rewrote the main clause (Line 475) for clear expression.
Line 475 – “This study proposes a fully 3D printed, monolithic soft robot capable of linear and turning locomotion.” – this paper is a review, which study?!
Answer: Thank you for finding this error. It is our luck to know about this error in reviewing, we apologize for this careless mistake, it is a true confusion.
Line 475 is the main clause to line 473. Here, this submitted manuscript is a typical review, not a research article as you pointed out.
We rewrote this main clause into “ The reviewed 4D laser-synthesized monolithic soft robots are capable of linear and turning locomotion at micro/nanoscale for breaking the limitations of volume and controllability by traditional chemical/physical methods.” in the conclusion part.
Figure 12 and the comparison are better fitted at the end of section “6. Advantages among current minimized 4D robots, challenges, and future” in my opinion.
Answer: We completely agree with your proper opinion, place of Figure 12 (changed to Figure 13, in reviewed versions) now is moved to section 6 in line with your suggestion. The corresponding description is slightly adjusted.
We believe this change will make sections 6 and conclusion more logically right and understandable by readers, thanks again for your opinion.
Comments on the Quality of English Language
Some phrases are unclear and should be revised!
Answer: Your emphasis on phase clarity is important and useful to our reviewing, we are sorry for the language and wording issues because our team is a pure Chinese domestic research team, and our English Language are inferior to native English speaker as you pointed out.
We acknowledge it is necessary to revise the unclear phrase. A series of language tool-based artificial intelligence, grammar tools (grammar, 1Checker) are introduced in revision to check the unsuitable phrases. Some confusing words and sentences are re-written for clarity.

Round 2
Reviewer 1 Report
Dear Authors,
Thank you very much considering my suggestions. All my concerns have been carefully answered and the necesseary modifications have been implemented in the manuscript. Therefore, I recommend it for publication. Congratulations for the great work.
Reviewer 3 Report
The authors have addressed all my reccomantadions, the quality of the paper improved, therefore, I reccomand it for publication.
The authors have addressed all my reccomantadions, the quality of the paper improved, therefore, I reccomand it for publication.